# Pharyngeal neuronal mechanisms governing sour taste perception in *Drosophila melanogaster*

**Bhanu Shrestha, Jiun Sang, Suman Rimal, Youngseok Lee***

Department of Bio & Fermentation Convergence Technology, Kookmin University, Seoul, Republic of Korea

## eLife Assessment

This is a **useful** contribution to our understanding of taste perception. The idea that specific receptors function in the pharynx to mediate responses to carboxylic acids is interesting, although the expression analysis is incomplete. Reviewers also have a number of other suggestions for improvement, including the request that authors provide more details about the methodology used. In general, the claims are supported by **solid** evidence and add to a growing body of literature on this topic.

***For correspondence:**
ylee@kookmin.ac.kr

**Competing interest:** The authors declare that no competing interests exist.

**Abstract** Sour taste, which is elicited by low pH, may serve to help animals distinguish appetitive from potentially harmful food sources. In all species studied to date, the attractiveness of oral acids is contingent on concentration. Many carboxylic acids are attractive at ecologically relevant concentrations but become aversive beyond some maximal concentration. Recent work found that *Drosophila* ionotropic receptors IR25a and IR76b expressed by sweet-responsive gustatory receptor neurons (GRNs) in the labellum, a peripheral gustatory organ, mediate appetitive feeding behaviors toward dilute carboxylic acids. Here, we disclose the existence of pharyngeal sensors in *Drosophila melanogaster* that detect ingested carboxylic acids and are also involved in the appetitive responses to carboxylic acids. These pharyngeal sensors rely on IR51b, IR94a, and IR94h, together with IR25a and IR76b, to drive responses to carboxylic acids. We then demonstrate that optogenetic activation of either *Ir94a*[+] or *Ir94h*[+] GRNs promotes an appetitive feeding response, confirming their contributions to appetitive feeding behavior. Our discovery of internal pharyngeal sour taste receptors opens up new avenues for investigating the internal sensation of tastants in insects.

## Introduction

Taste perception is a crucial sensory modality for insects, as it allows them to identify and respond to food sources, potential toxins, and mating partners. In *Drosophila*, the molecular and neural mechanisms underlying taste perception have been extensively studied, revealing a complex and highly specialized system for detecting different tastants (*Montell, 2021*; *Shrestha and Lee, 2023b*). Sour taste has been shown to play an important role in determining feeding preferences and mediating aversive responses to noxious compounds. Sour taste is evoked by the hydrogen ion (H[+]) in acidic compounds, regulating dietary intake and pH balance. In mammals, low pH is detected by proton channels like otopetrin 1 (Otop1) (*Teng et al., 2019*; *Zhang et al., 2019*). In *Drosophila*, otopetrin-like A (OtopLA), a conserved proton channel, is proposed to be necessary for acid sensitivity (*Ganguly et al., 2021*; *Mi et al., 2021*). Previous research indicates that taste perception of sourness involves protons, anions, and concentration (*Rimal et al., 2019*).

Insects employ various sensory mechanisms to detect chemicals in their environment. Olfactory organs (antennae and maxillary palps) detect volatile chemicals, while non-volatile chemicals are sensed through gustatory organs like labella, legs, wing margins, and ovipositors. The labellar hairs and pegs play a crucial role in feeding. Each hemisphere of the labellum contains 31 sensilla, categorized by length into long (L-type), intermediate (I-type), and short types (S-type). These sensilla house gustatory receptor neurons (GRNs), mechanosensory neurons, and supporting cells. Several chemosensory receptors, including ionotropic receptors (IRs), pickpocket (PPK) ion channels, transient receptor potential (TRP) ion channels, and gustatory receptors (GRs), facilitate chemical detection. Receptors like alkaliphile (alka) and OTOP1 are involved in alkaline taste sensation in *Drosophila* and vertebrates, respectively (*Mi et al., 2023*; *Tian et al., 2023*). GRs and IRs are associated with the perception of various chemicals, with specific GRs and IRs identified for different compounds like sugars, carboxylic acids, vitamins, and metals (*Dahanukar et al., 2007*; *Li et al., 2023*; *Luo et al., 2022*; *Shrestha et al., 2023a*; *Shrestha and Lee, 2021*; *Stanley et al., 2021*; *Xiao et al., 2022*). For example, IR25a and IR76b have recently been identified as coreceptors for taste identification, playing a novel role in perceiving salts such as sodium and calcium (*Jaeger et al., 2018*; *Lee et al., 2018*; *Zhang et al., 2013*), metals such as zinc and other heavy metals (*Xiao et al., 2022*), amino acids (*Aryal et al., 2022a*; *Ganguly et al., 2017*), carboxylic acid and fatty acids (*Ahn et al., 2017*; *Chen and Amrein, 2017a*), and vitamin C (*Shrestha et al., 2023a*). IR7a, originally known as an acetic acid sensor, has been discovered to function as a cadmium sensor, whereas IR47a acts independently of bitter GRNs in the perception of heavy metals (*Li et al., 2023*). IR56b, expressed in ppk23$^+$ neurons, governs aversion to Zn$^{2+}$ (*Luo et al., 2022*) and is also involved in the perception of low salt from sweet-sensing GRN (*Dweck et al., 2022*). Furthermore, recent studies have characterized IR7c as a high-salt sensor (*McDowell et al., 2022*), IR56d as a low-hexanoic acid sensor (*Ahn et al., 2017*; *Brown et al., 2021*; *Pradhan et al., 2023*), and IR25a and IR76b in the labellum as essential to detect attractive carboxylic acids (*Shrestha and Lee, 2021*; *Stanley et al., 2021*).

In *Drosophila*, ingested food first contacts dendritic processes in the taste sensilla in the pharynx. The axons project to distinct parts of the subesophageal zone (SEZ) responsible for taste sensation in the fly's brain. Despite their importance, few studies have explored the physiological responses of pharyngeal GRNs (*Chen and Dahanukar, 2017b*; *Chen et al., 2021*; *Chen et al., 2019*). Here, we focus on investigating the molecular mechanism of attractive acid taste in *Drosophila*. Lactic acid (LA), citric acid (CA), and glycolic acid (GA) are used as test compounds in behavioral screens. LA is known to be palatable and to extend lifespan, whereas CA and GA are energy sources found in citrus fruits (*Massie and Williams, 1979*; *Rabinowitz and Enerbäck, 2020*; *Rimal et al., 2019*; *Stanley et al., 2021*). Behavioral and physiological assays have shown that sweet-sensing GRNs on the labellum mediate the appetitive preference of *Drosophila melanogaster* toward low concentrations of carboxylic acids (*Shrestha and Lee, 2021*; *Stanley et al., 2021*). Here, we show that mutations in pharyngeal IRs result in reduced preference or aversion to LA, CA, and GA, indicating the essential role of these IRs in mediating the attraction to low acid concentrations. Peripheral (both labellar and pharyngeal) taste sensation of all three acids relies on broadly tuned IR25a and IR76b receptors, but IR51b is essential for their internal perception. Additionally, IR94a and IR94h are also essential for pharyngeal perception of CA and GA in the pharynx. These findings highlight a unique mechanism of sour taste perception, where external and internal organs have distinct roles in sensing acid taste.

## Results

### IRs are indispensable for attracting organic carboxylic acids

In our previous study, we observed that control flies exhibited attraction toward food containing 1% carboxylic acids, indicating their ability to detect sour taste through specific taste receptors, but we did not identify the complete receptors (*Shrestha and Lee, 2021*). Here, to identify the receptors responsible, we conducted a screening of various *Ir* mutants, targeting receptors with known functional roles as well as those yet to be identified (*Figure 1*). We dissolved the tastants in 72-well microtiter plates featuring an alternating pattern of 2 mM sucrose alone or combined with 1% LA, 1% CA, or 1% GA. The mutant library consisted of broadly expressed receptors *Ir25a* and *Ir76b*, as well as specific sensors for acetic acid (*Ir7a*) (*Rimal et al., 2019*), high salt (*Ir7c*) (*McDowell et al., 2022*), cantharidin (*Ir7g*, *Ir51b*, and *Ir94f*) (*Pradhan et al., 2024*), low salt (*Ir56b*) (*Dweck et al., 2022*), fatty

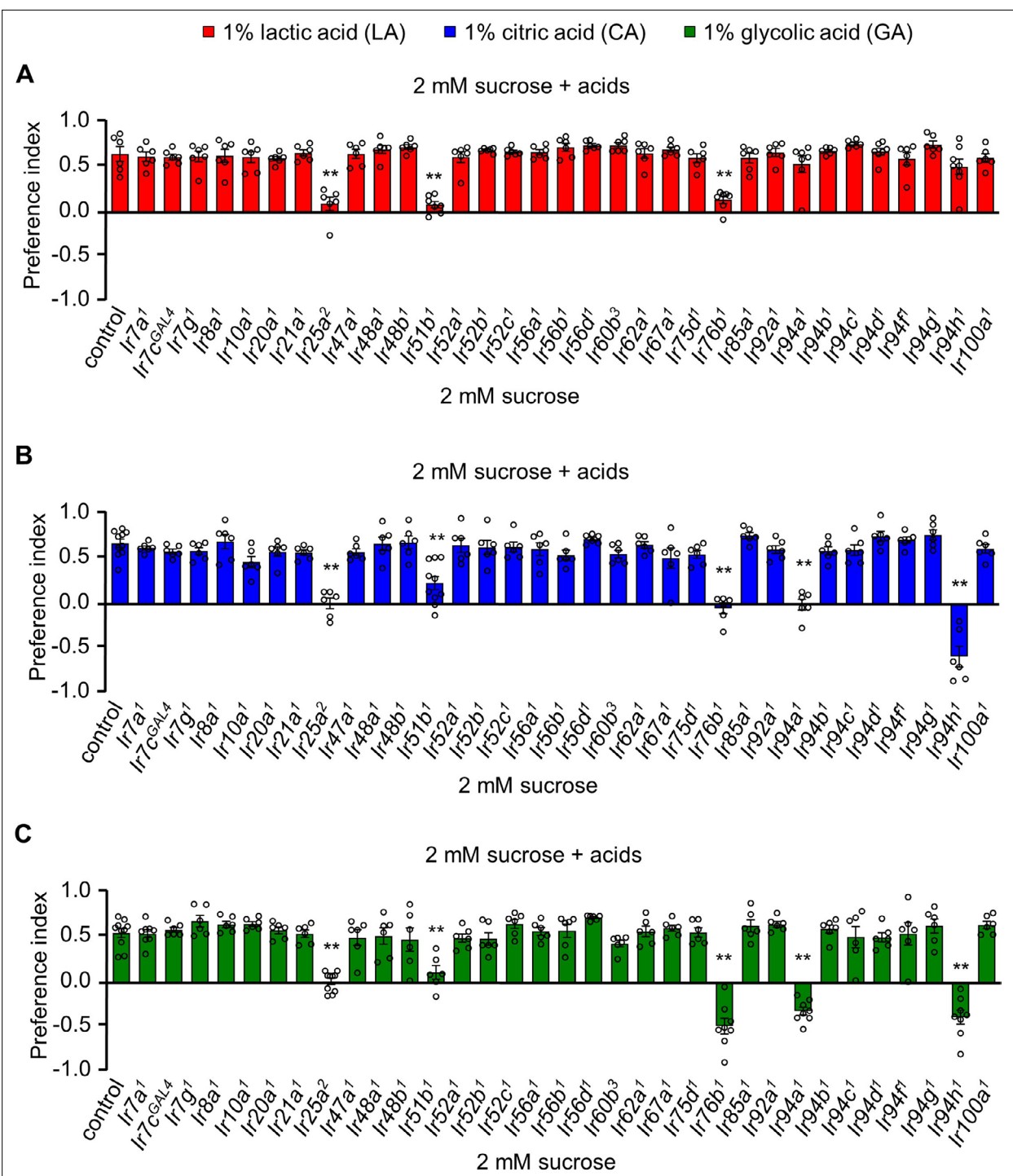

**Figure 1.** Ionotropic receptors are essential for sensing carboxylic acids in *Drosophila melanogaster*. Screening of candidate 33 *Ir* mutants through binary food choice assay in presence of (**A**) 1% lactic acid (LA), (**B**) 1% citric acid (CA), and (**C**) 1% glycolic acid (GA) (*n* = 6–8 biological replicates). The control flies were $w^{1118}$. All error bars represent SEMs. Single-factor ANOVA with Scheffe's analysis was used as a post hoc test to compare multiple sets of data. Circular dots represent the number of trials performed. Asterisks indicate statistical significance compared with control; **$p < 0.01$.

The online version of this article includes the following figure supplement(s) for figure 1:

**Figure supplement 1.** Structural analysis of (**A**) *Ir94a[1]* and (**B**) *Ir94h[1]* genes, (**C**) gender-dependent response to 1% carboxylic acid.

acids (*Ir56d*) (*Ahn et al., 2017*), nitrogenous waste and aversive amino acids (*Ir51b*) (*Aryal et al., 2022a*; *Dhakal et al., 2021*), Ca$^{2+}$ mineral (*Ir62a*) (*Lee et al., 2018*), and alkaline taste (*Ir20a*, *Ir47a*, *Ir51b*, *Ir52a*, *Ir92a*, and *Ir94f*) (*Pandey et al., 2024*). *Ir8a* was also included in the screening list as it is known for mediating LA olfactory attraction in *Aedes aegypti* and avoidance in *D. melanogaster* (*Ai et al., 2013*; *Raji et al., 2019*). Among the defective *Ir* lines, most showed a preference for LA, CA, and GA, but the response of *Ir25a²*, *Ir51b¹*, and *Ir76b¹* to 1% LA, CA, and GA was significantly reduced compared to control flies (*Figure 1A—C*). Furthermore, the *Ir94a¹* and *Ir94h¹* mutants, generated through ends-out homologous recombination (*Figure 1—figure supplement 1A, B*), exhibited lower preference or aversion to CA and GA (*Figure 1B, C*). These receptors (IR51b, IR94a, and IR94h) might assemble into multimeric complexes, with IR25a and IR76b serving as essential, yet general, subunits. The IR25a and IR76b are supplemented by additional subunits such as IR51b, IR94a, and IR94h that confer specificity for carboxylic acid. The functional role of IR94a and IR94h in behavior and physiology was unknown, although their expression was observed in the pharyngeal region of both larvae and adult flies (*Chen and Dahanukar, 2017b*; *Stewart et al., 2015*). The *Ir94h¹* mutant showed severe defects in the attraction to CA and GA, and similarly, *Ir76b¹* and *Ir94a¹* mutants to GA compared to other candidate mutants. Overall, we suggest that five IRs (*Ir25a*, *Ir51b*, *Ir76b*, *Ir94a*, and *Ir94h*) exhibit strong potential as receptors for sour taste.

Most insects undergo profound physiological and behavioral changes during mating (*Wolfner et al., 2005*). Previous studies in *Drosophila* show that after mating, females change their food preferences in addition to increasing their feeding activities (*Ribeiro and Dickson, 2010*; *Vargas et al., 2010*). Moreover, many animals exhibit a dietary shift in response to changes in environmental or developmental requirements (*Waldbauer and Friedman, 1991*). Motivated by these results, we set out to investigate if changes in mating status are associated with a shift toward a preference for sour taste and, if so, whether this preference is dependent on the sex of the fly. Therefore, we conducted experiments to investigate potential sex differences and the effect of mating state on the attraction to LA, CA, and GA. We found that the attraction to the carboxylic acids was indistinguishable among mated male flies, virgin female flies, and mated female flies (*Figure 1—figure supplement 1C*). These results suggest that sex and mating status do not significantly impact the flies' response to carboxylic acid in terms of attraction, and the physiological and developmental status of the organism are not important factors for nutritional assessment of carboxylic acid value. After establishing that sour taste perception is not sex-specific, we proceed to validate potential receptors for the sensation of sour taste. Thus to determine the function of *Ir51b* in detecting carboxylic acids, we examined an additional mutant, *Ir51b²* (*Dhakal et al., 2021*). This mutant exhibited a similar deficit as *Ir51b¹*, suggesting the essential role of *Ir51b* in GRN-mediated sour taste perception (*Figure 2A*). Previously, the broadly expressed IRs, IR25a and IR76b, were identified as key molecular sensors for LA, CA, and GA in the peripheral region (*Shrestha and Lee, 2021*). Having identified *Ir51b* as a key sensor for carboxylic acid, we conducted a recovery experiment. We used the broadly expressed *Ir25a-GAL4* to drive *UAS-Ir51b* in the mutant background. The responses to LA, CA, and GA were fully rescued at a concentration of 1% (*Figure 2B*). In contrast, the feeding behavior of the parental lines phenocopied that of defective *Ir51b¹*. Overall, these findings demonstrate that *Ir51b* is required for carboxylic acid sensing.

Next, we examined the effect of *Ir94a¹* and *Ir94h¹* combined with deficiencies (*Df*) that uncover the respective genes. We observed that the defect persisted in the deficiency lines (*Figure 2C*). After observing the defect in *Ir94a¹* and *Ir94h¹* flies in sensing CA and GA in feeding behavior assays, we performed rescue experiments using the respective *GAL4* drivers. We found that the mutant phenotype of *Ir94a¹* and *Ir94h¹* flies was fully restored to normal by expressing wild-type *Ir94a* and *Ir94h* transgenes, respectively (*Figure 2D, E*). By rescuing the deficits of *Ir94a¹* and *Ir94h¹* using the respective *GAL4* lines, we confirmed that *Ir94a* and *Ir94h* are essential for the perception of carboxylic acid-mediated sensation.

Previous research showed that acetic acid detection by IR7a was contingent on pH and concentration (*Rimal et al., 2019*). Therefore, we conducted a series of experiments to investigate the behavioral responses of fruit flies to different concentrations and pH ranges of carboxylic acids. Firstly, we performed feeding assays with and without the presence of sucrose to determine the influence of sugar on flies' behavior (*Figure 2—figure supplement 1A, B*). Control flies ($w^{1118}$) exhibited attraction to 0.5% and 1% carboxylic acid. In contrast, at 5% and 10% LA, the attraction was relatively lower, and flies rather avoided 10% CA and 5% and 10% GA. All candidate mutants displayed significant

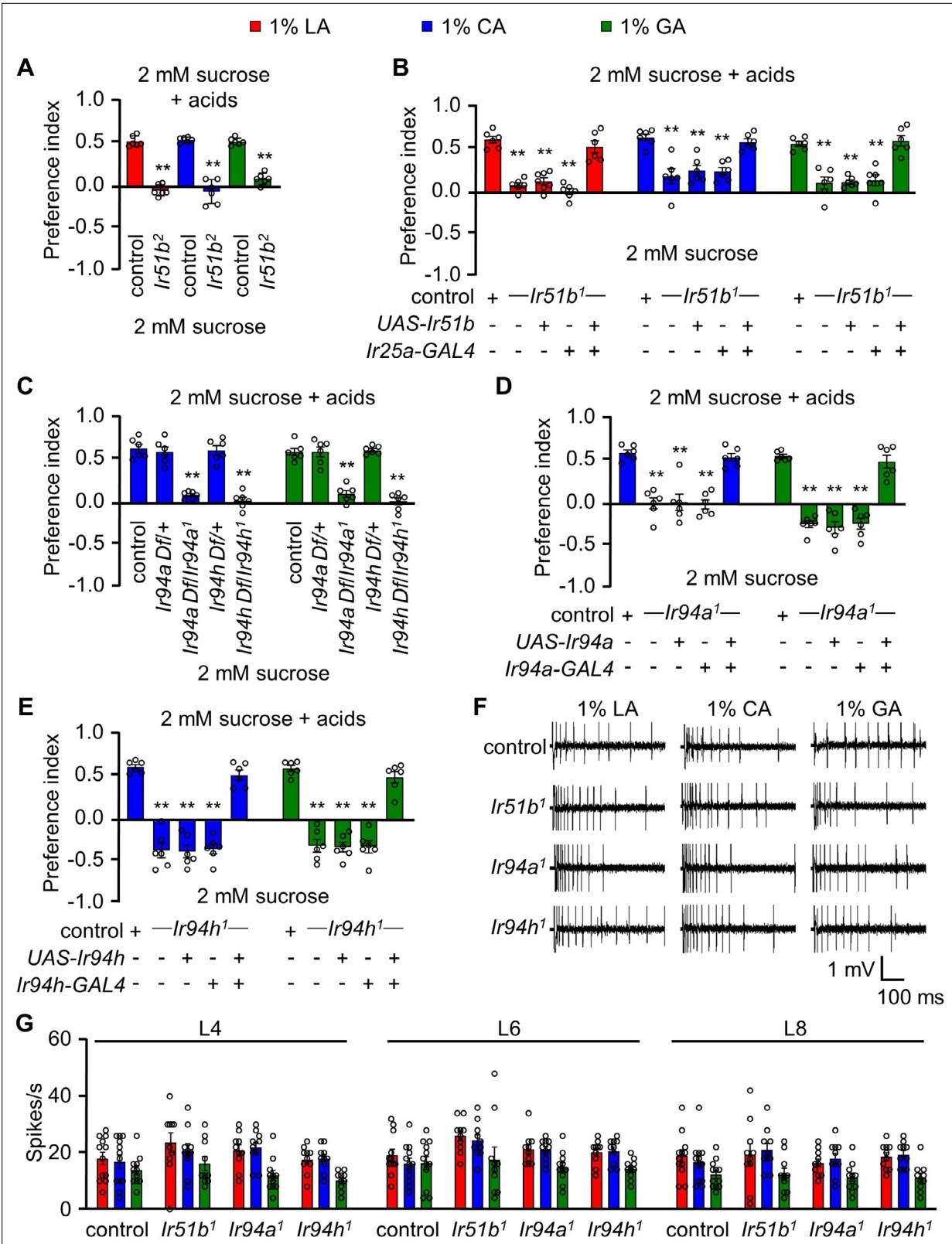

**Figure 2.** IR51b, IR94a, and IR94h are required to sense 1% lactic acid (LA), citric acid (CA), and glycolic acid (GA). (**A**) Two-way choice assay of control and *Ir51b²* (in-frame knock-in of GAL4) in 1% LA, CA, and GA (n = 6 biological replicates). (**B**) Rescue of *Ir51b¹* mutant defect to sense the taste of carboxylic acid in binary food choice assay driven under the control of broadly expressed *Ir25a-GAL4* (n = 6–10 biological replicates). (**C**) Feeding assay analysis of *Ir94a* and *Ir94h* mutation in trans with a deficiency (Df) that spans *Ir94a* and *Ir94h*, respectively, in 1% CA and GA (n = 6 biological replicates).

*Figure 2 continued on next page*

*Figure 2 continued*

(**D**) Binary food choice assay to rescue the defect of *Ir94a¹* by expressing wild-type cDNA under the control of *Ir94a-GAL4* in presence of 1% CA and GA (n = 6–8 biological replicates). (**E**) Rescuing the defect of *Ir94h¹* by expressing wild-type cDNA under the control of *Ir94h-GAL4* in presence of 1% CA and GA (n = 6–8 biological replicates). (**F**) Sample trace representing the action potential from 1% CA in control, *Ir51b¹*, *Ir94a¹*, and *Ir94h¹* flies. (**G**) Average number of spikes per second of control flies along with *Ir51b¹*, *Ir94a¹*, and *Ir94h¹* flies stimulated by 1% LA, CA, and GA from L-type sensilla (L4, L6, and L8) (n = 10 biological replicates). The control flies were *w¹¹¹⁸*. All error bars represent SEMs. Single-factor ANOVA with Scheffe's analysis was used as a post hoc test to compare multiple sets of data. Circular dots represent the number of trials performed. Black asterisks indicate statistical significance compared with control; **p < 0.01.

The online version of this article includes the following figure supplement(s) for figure 2:

**Figure supplement 1.** Carboxylic acid profiling in the (**A**) presence or (**B**) absence of varying sucrose concentrations and (**C**) pH levels.

defects in attraction to 0.5% and 1% of the tested carboxylic acid food. The difference in preference between control flies and mutants was particularly pronounced at 1% concentration for all three of the tested acids (*Figure 2—figure supplement 1A, B* and *Supplementary file 1*, *Supplementary file 2*). This implies that the ideal concentration to attract flies is 1% carboxylic acid. We thus proceeded with further studies using a 1% concentration. Interestingly, *Ir94a¹* and *Ir94h¹* showed normal responses to all concentrations of LA. The absence of sugar removes the competition between the activation of avoidance and attraction sensory neurons, revealing the role of the tastant itself present in the food. However, the reduction in preference or avoidance behavior persisted in the candidate mutants, both in the presence and absence of sucrose, at concentrations ranging from 0.1% to 5%. Intriguingly, the response toward 10% carboxylic acid food was similar in control flies and mutants. These results indicate the involvement of the five identified IRs in sensing low concentrations of carboxylic acid in the context of sour taste.

Feeding assays were performed using carboxylic acids with varying pH levels, from acidic to neutral, to examine the effect of pH on the behavioral characteristics of control flies and candidate IR mutants. The results revealed that pH influenced the behavioral responses of both control flies and mutants, with a higher preference observed for low-pH acids compared to neutral pH (*Figure 2—figure supplement 1* and *Supplementary file 3*). The pH effects were distinguishable among control flies and mutants in the acidic range of approximately pH 2.3–2.7. Even after adjusting the pH to 3.5 using KOH, the contrasting behavior between control flies and mutants persisted. In contrast, as the pH increased to approximately pH 5.0–7.0, the behavior became similar between control flies and mutants. These findings indicate that the pH of the acid is a significant factor in taste discrimination. Collectively, these data demonstrate that the pH, concentration, and chemical structure of the acid influence attraction to sour taste.

Electrophysiological techniques offer a potent means to investigate the peripheral taste response of insects, complementing behavioral assays. Through electrophysiological assays, we can directly and quantitatively measure peripheral gustatory responses. We previously identified the role of IR25a and IR76b for the sensation of carboxylic acids from labellum using tip recording assay (*Shrestha and Lee, 2021*). To investigate whether newly identified *Ir* mutants elicit abnormal neuronal responses from the labellum, we conducted tip recording assays to assess the electrophysiological response of *Ir51b¹*, *Ir94a¹*, and *Ir94h¹* in L-type sensilla (L4, L6, and L8). We observed normal neuronal firing in all three mutants when stimulated with 1% LA, CA, and GA (*Figure 2F, G*). Based on the results of the binary food choice assay and tip recording assay, we hypothesize that *Ir51b*, *Ir94a*, and *Ir94h* may play a role in the sensation of carboxylic acid in pharyngeal organs.

## *Ir51b*, *Ir94a*, and *Ir94h* exhibit distinct patterns of expression in the pharynx

Because the behavioral defects of *Ir94a¹* and *Ir94h¹* mutant flies to detect CA and GA were fully rescued by their GAL4 lines, respectively, we expanded our research to investigate the expression pattern of these receptors using their *GAL4* drivers. Within the adult fly pharynx, three distinct pharyngeal taste organs can be distinguished: the labral sense organ (LSO), as well as the ventral and dorsal cibarial sense organs (VCSO and DCSO). The VCSO and DCSO are located on opposite sides of the rostrum, whereas the LSO is situated in the haustellum (*Figure 3A*). *Ir94a* was expressed in the VCSO region of the pharynx (*Figure 3B*), whereas *Ir94h* showed neuronal expression in both the LSO and VCSO regions (*Figure 3C, D*). This analysis, based on the location of expression, provides

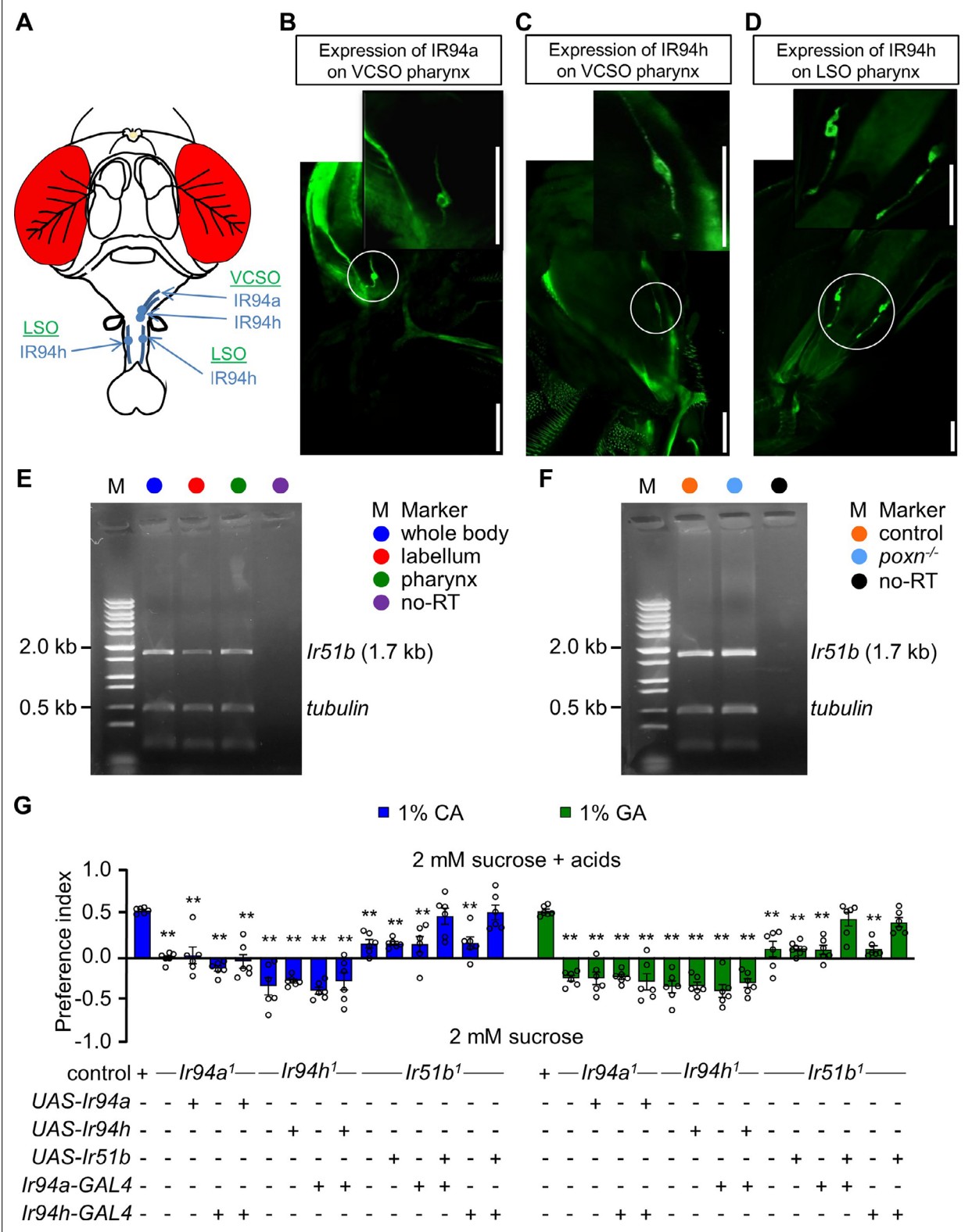

**Figure 3.** *Ir51b*, *Ir94a*, and *Ir94h* are expressed in pharynx of *Drosophila melanogaster*. (**A**) Schematic diagrammatic representation of head of *Drosophila* depicting the location of *Ir94a* and *Ir94h* pharyngeal neuron. Images showing the expression of *Ir94a* and *Ir94h* neuron in different regions of pharynx, (**B**) the expression of *Ir94a-GAL4* in ventral cibarial sense organ (VCSO) region of pharynx, (**C**) the expression of *Ir94h-GAL4* in VCSO region of pharynx, and (**D**) the expression of *Ir94h-GAL4* in labral sense organ (LSO) region of pharynx (scale bars, 100 μm). (**E**) Gel picture of RT-PCR resulting *Ir51b* expression in whole body, labellum, and pharynx. In lanes designated as 'no-RT', polyA+ RNA extracts underwent RT omission, and the absence

*Figure 3 continued on next page*

*Figure 3 continued*

of PCR products serves as evidence of the absence of genomic DNA contamination. Amplified tubulin (0.5 kb) was used as control. DNA ladder marker is denoted by 'M'. (**F**) Gel picture of results from RT-PCR showing *Ir51b* expression in control (*w*[1118]) and *poxn* null alleles (*poxn*[70] and *poxn*[ΔM22−B5]) from pharynx. In lanes designated as 'no-RT', polyA+ RNA extracts underwent RT omission, and the absence of PCR products serves as evidence of the absence of genomic DNA contamination. Amplified tubulin (0.5 kb) was used as control. DNA ladder marker is denoted by 'M'. (**G**) Feeding assay to rescue the defect of *Ir94a*[1] and *Ir51b*[1] by *Ir94h-GAL4*; and *Ir94h*[1] and *Ir51b*[1] by *Ir94a-GAL4* in presence of 1% citric acid (CA) and glycolic acid (GA) (*n* = 6 biological replicates). All error bars represent SEMs. Single-factor ANOVA with Scheffe's analysis was used as a post hoc test to compare multiple sets of data. Circular dots represent the number of trials performed. Black asterisks indicate statistical significance compared with control; **p < 0.01.

The online version of this article includes the following source data and figure supplement(s) for figure 3:

**Source data 1.** Original gel picture to *Figure 3*, panel E.

**Source data 2.** Original gel picture to *Figure 3*, panel F.

**Figure supplement 1.** Limited expression of *Ir94a* and *Ir94h* in (**A-C**) gustatory organs and (**D**) brain suboesophageal region.

---

evidence that *Ir94a* and *Ir94h* are distinct neurons within the pharynx. Additionally, we previously generated a knock-in-GAL4 line for *Ir51b* (referred to as *Ir51b*[2]). However, we were unable to visualize the expression of *Ir51b* using a green fluorescent protein (GFP) reporter. Nevertheless, we successfully recapitulated the expression of *Ir51b* in the bitter-sensing GRNs of the labellum using RT-PCR (***Dhakal et al., 2021***). Considering the broad expression of *Ir51b*, including the GRNs involved in the attractive sensing of low levels of acids in the pharynx, as demonstrated by the defect in sensing low concentrations of carboxylic acids through feeding assays in *Ir51b* mutants, we sought to validate this speculation. We performed RT-PCR analysis on cDNA extracted from control and *poxn* mutant flies which exhibit alteration of chemosensory bristles to mechanosensory bristles at labellar organs (***Boll and Noll, 2002***). The fact that expression was also detected in the pharynx provided further support for our previous hypothesis (***Figure 3E, F*** and ***Figure 3—source data 1***, ***Figure 3—source data 2***).

To further investigate the expression of *Ir94a* and *Ir94h* chemosensory neurons, we examined various other gustatory organs, including peripheral olfactory organs (antennae and maxillary palps), forelegs, and the labellum. However, we did not observe neuronal expression in these different chemosensory body parts (***Figure 3—figure supplement 1A, B***). Furthermore, the axonal projection of *Ir94a* and *Ir94h* GRNs to the SEZ, which is a brain region responsible for processing taste signals (***Figure 3—figure supplement 1C, D***), suggests that IR94a and IR94h play a role in taste perception from the pharynx. However, because both *Ir94a* and *Ir94h* neurons exhibit defects in sensing the same compounds, we were curious about the potential association between them at the level of neural circuit connections. To this end, we conducted a rescue experiment using *Ir94a-GAL4* and *Ir94h-GAL4* to rescue the defects of *Ir94h*[1] mutant and *Ir94a*[1] mutant, respectively (***Figure 3G***). This genetic experiment demonstrated a failure in rescuing the defects, indicating no association between the roles of IR94a and IR94h in sensing sour taste. *Ir94a* and *Ir94h* appear to operate independently in the perception of sour taste, with no complementary or redundant functions between them in this specific sensory process. Using the pharyngeal-specific neurons *Ir94a-GAL4* and *Ir94h-GAL4*, we attempted to rescue the defect observed in *Ir51b*[1], and intriguingly, we found that both pharyngeal neurons successfully rescued the defect (***Figure 3G***). Because *Ir94a* and *Ir94h* neurons occupy different locations within the pharynx, this finding led us to speculate that the activation of a single neuron is adequate to confer the activity of IR51b.

## Functional validation of pharyngeal *Ir94a* and *Ir94h* GRNs in detecting appealing sour taste

To investigate the role of pharyngeal neurons in *Drosophila* in relation to the pharyngeal sour sensation, we examined flies *trans*-heterozygous for two *poxn* null alleles (*poxn*[70] and *poxn*[ΔM22−B5]) (***Boll and Noll, 2002***). We chose to use *poxn* mutants in our study due to their suitability for studying pharyngeal taste coding, as they lack taste function in the legs and labial palps but retain intact pharyngeal sweet taste, which is necessary and sufficient for driving the preferred consumption of sweet compounds through prolonged ingestion. We assessed the behavior of *poxn* mutant flies using pre-ingestion contact analysis (proboscis extension response [PER] assay) and post-ingestion contact analysis (binary food choice assay). Our findings revealed that there was no attractive response in the PER assay, but prolonged consumption showed increased feeding of low concentrations of acids (***Figure 4A, B***). This

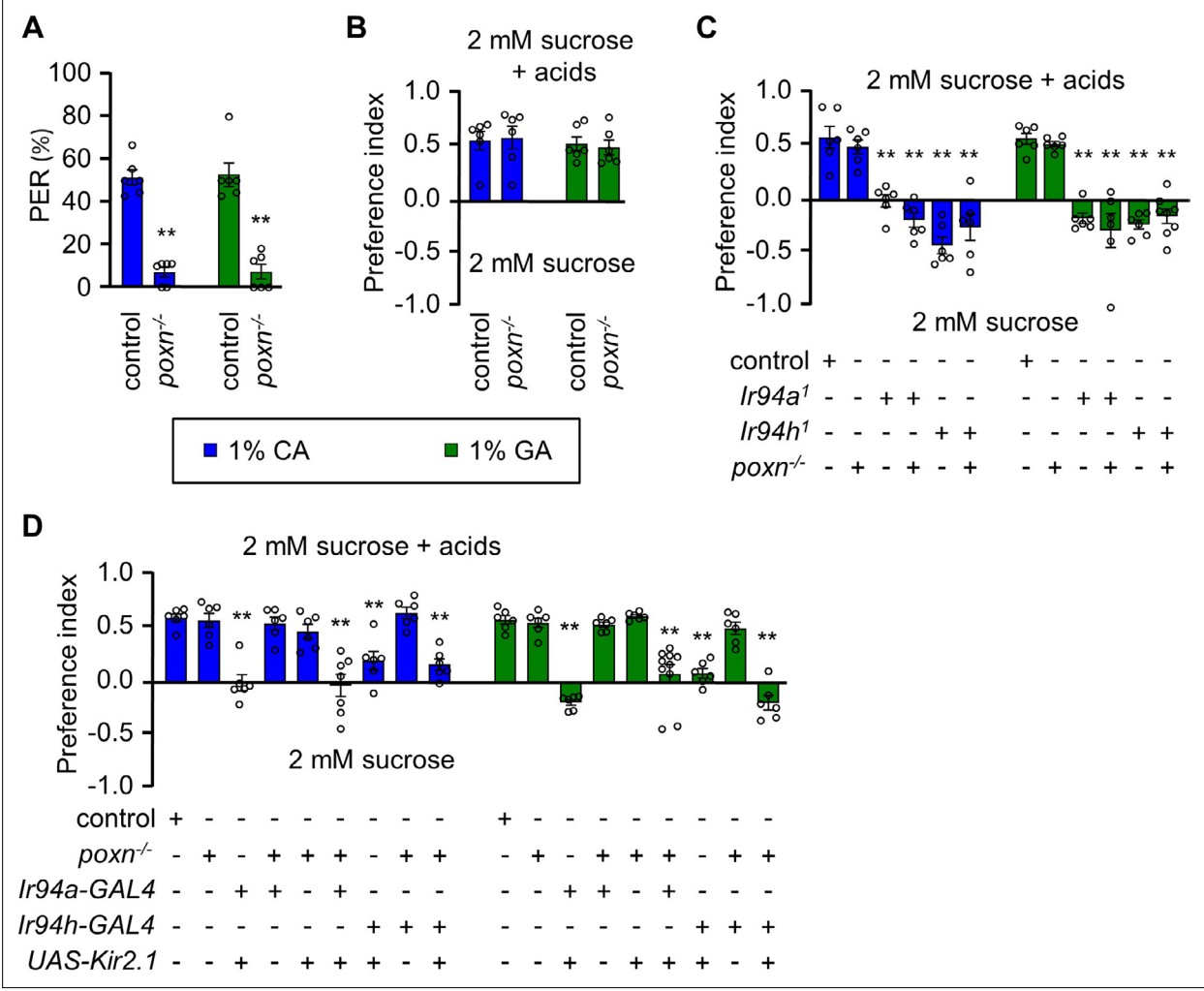

**Figure 4.** Pharyngeal gustatory receptor neurons (GRNs) have a role in sensing the taste of carboxylic acids. (**A**) Proboscis extension response (PER) assay, (**B**) Binary food choice assay of control and *poxn*−/− mutant (*poxn70/poxnΔM2−−B5*) in presence of 1% citric acid (CA) and 1% glycolic acid (GA) (n = 6 biological replicates). (**C**) Feeding assay control and mutant line (*Ir94a1* and *Ir94h1*) in *poxn*−/− mutant (*poxn70/poxnΔM22−B5*) mutant background in 1% CA and 1% GA (n = 6–8 biological replicates). (**D**) Mean preference index from binary food choice assay in presence of 1% CA and 1% GA for genetically silenced flies. Genetic manipulations were performed in *poxn*−/− mutant (*poxn70/poxnΔM22−B5*) mutant background for *Ir94a-GAL4* and *Ir94h-GAL4* by crossing with *UAS-Kir2.1* (n = 6–10 biological replicates). The control flies were *w1118*. All error bars represent SEMs. Single-factor ANOVA with Scheffe's analysis was used as a post hoc test to compare multiple sets of data. Black circular dots represent the number of trials performed. Black asterisks indicate statistical significance compared with control; **p < 0.01.

observation may be attributed to the presence of functional pharyngeal neurons that drive attraction to sour stimuli in *poxn* mutants, whereas labellar attractive sour taste sensation is apparently absent (*Chen and Dahanukar, 2017b*; *Chen et al., 2019*). As alluded to above, pharyngeal neurons are activated prior to post-ingestion, indicating a similar response during feeding. Therefore, these results indicate that the pharyngeal neurons driving feeding attraction to carboxylic acids remain intact in the tested *poxn* mutants.

Next, we investigated the role of *Ir94a1* and *Ir94h1* in the context of feeding attraction in the *poxn* mutant background. We discovered that flies lacking peripheral sensory receptors in presence of *Ir94a1* and *Ir94h1*, exhibited defects in sensing attractive taste, leading to a reduction in attractiveness to carboxylic acids. The presence of *Ir94a1* and *Ir94h1* influenced this defect, but the *poxn* mutant itself did not (*Figure 4C*). Similar results were observed for both the CA and GA conditions. Furthermore, we suppressed pharyngeal gene regulatory networks in the *poxn* mutant background by expressing the inward rectifying potassium channel Kir2.1 under the control of *Ir94a-GAL4* and *Ir94h-GAL4*. Interestingly, flies with silenced *Ir94a-GAL4* and *Ir94h-GAL4* but with intact *poxn* mutant genes lost

their capacity for attraction to appealing concentrations of acid (*Figure 4D*). This suggests that *poxn* mutants lacking *Ir94a* and *Ir94h* pharyngeal GRNs may be unable to identify attractive concentrations of CA and GA and that the sensation of attractive concentrations may involve the operation of a post-ingestive mechanism under the influence of pharyngeal *Ir94a-GAL4* and *Ir94h-GAL4*.

## Artificial activation of pharyngeal *Ir94a* and *Ir94h* GRNs by chemogenetics and optogenetics

We wanted to gain further insights into the functionality of *Ir94a* and *Ir94h* GRNs, so we conducted experiments to artificially activate each GRN by expressing rat *trpV1* and feeding capsaicin. The flies were given a choice between consuming 2 mM sucrose or 2 mM sucrose plus 100 µM capsaicin. Control flies expressing only *Ir94a-GAL4* and *Ir94h-GAL4* or *UAS-trpV1* (*Marella et al., 2006*) genes displayed almost no bias to capsaicin (*Figure 5—figure supplement 1*). In contrast, flies expressing *UAS-trpV1* under the control of *Ir94a-GAL4* and *Ir94h-GAL4* GRNs exhibited attraction to capsaicin, indicating that the activation of *Ir94a-GAL4* and *Ir94h-GAL4* GRNs leads to gustatory attraction (*Figure 5—figure supplement 1*). Conversely, flies expressing *UAS-trpV1* under the control of *ppk23-GAL4* (calcium-sensing GRNs) avoided capsaicin, demonstrating that the activation of *ppk23* GRNs induces gustatory avoidance. Additionally, the activation of *Gr64f-GAL4* (sweet-sensing GRNs) promoted capsaicin attraction (*Figure 5—figure supplement 1*).

To further investigate the optogenetic activation of *Ir94a* and *Ir94h* GRNs, we expressed the red-light activated channelrhodopsin CsChrimson (*Klapoetke et al., 2014*) in these neurons (*Figure 5A–C*). Flies were reared on standard food with all-*trans*-retinal (ATR) and then starved for 18 hr before inducing PER using red light (*Figure 5A*). We first confirmed that the activation of *Gr5a* GRNs (sweet-sensing GRNs) with a red-light source alone, but not white light, resulted in the stimulation of PER, whereas the activation of *Gr66a* GRNs (bitter-sensing GRNs) had no effect (*Figure 5B*). Next, we discovered that stimulating the pharyngeal *Ir94a* and *Ir94h* GRNs induced PER at a similar level to sweet taste neuron activation (*Figure 5B*). Overall, these results conclude that *Ir94a* and *Ir94h* neurons are attraction-mediating neurons.

We also investigated whether *Ir94a* and *Ir94h* neurons could elicit any aversive response (*Figure 5C*). To this end, we conducted a PER assay to assess the response of the indicated flies to sucrose stimuli under two conditions: without red light (light off) and with red light (light on). The positive extension response of the proboscis persisted in the presence of strong red light in all control flies as well as *Gr5a* GRN-activated flies but not *Gr66a* GRN-activated flies. The PER persisted in *Ir94a* and *Ir94h* GRNs, similar to the response observed in *Gr5a* GRN. This indicates that *Ir94a* and *Ir94h* neurons are not aversion-mediating GRNs.

## CA and GA physiologically activate pharyngeal *Ir94a* and *Ir94h* GRNs

Our next hypothesis aimed to investigate the physiological role of IR94a and IR94h pharyngeal GRNs in responding to carboxylic acids. To achieve this, we expressed the calcium indicator GCaMP6s in *Ir94a-GAL4* and *Ir94h-GAL4* neurons and stimulated their expression using tastants. We then conducted ex vivo calcium imaging in the pharynx to measure changes in fluorescence intensity in the neurons.

Remarkably, we observed a gradual increase in the relative intensity level of fluorescence upon application of 1% CA and 1% GA (*Figure 6A—D* and *Videos 1 and 2*), indicating robust activation of the pharyngeal GRNs in response to these stimuli. However, we noted that the GCaMP6 signal decreased below the pre-stimulus baseline of neuronal fluorescent activity after a certain period of time, potentially due to continuous exposure to the CA and GA in our experimental setup. Upon quantification, we found higher fluorescence activity in control flies compared to mutant background flies, which exhibited reduced activity in response to CA and GA (*Figure 6B, D*). However, the reduced fluorescence activity in the mutant background was restored when driven by cell type-specific expression of the respective genes (*Figure 6A—D*). We then examined the physiological response to lower concentrations of the carboxylic acids to analyze whether minor concentrations were sufficient to elicit neuronal activity. We applied 0.1% CA and GA and observed changes in neuronal activity (*Figure 6E*). We observed a comparatively lower neuronal response compared to 1% CA and GA, indicating that the optimal appetitive concentration elicits a stronger physiological response (*Figure 6E*).

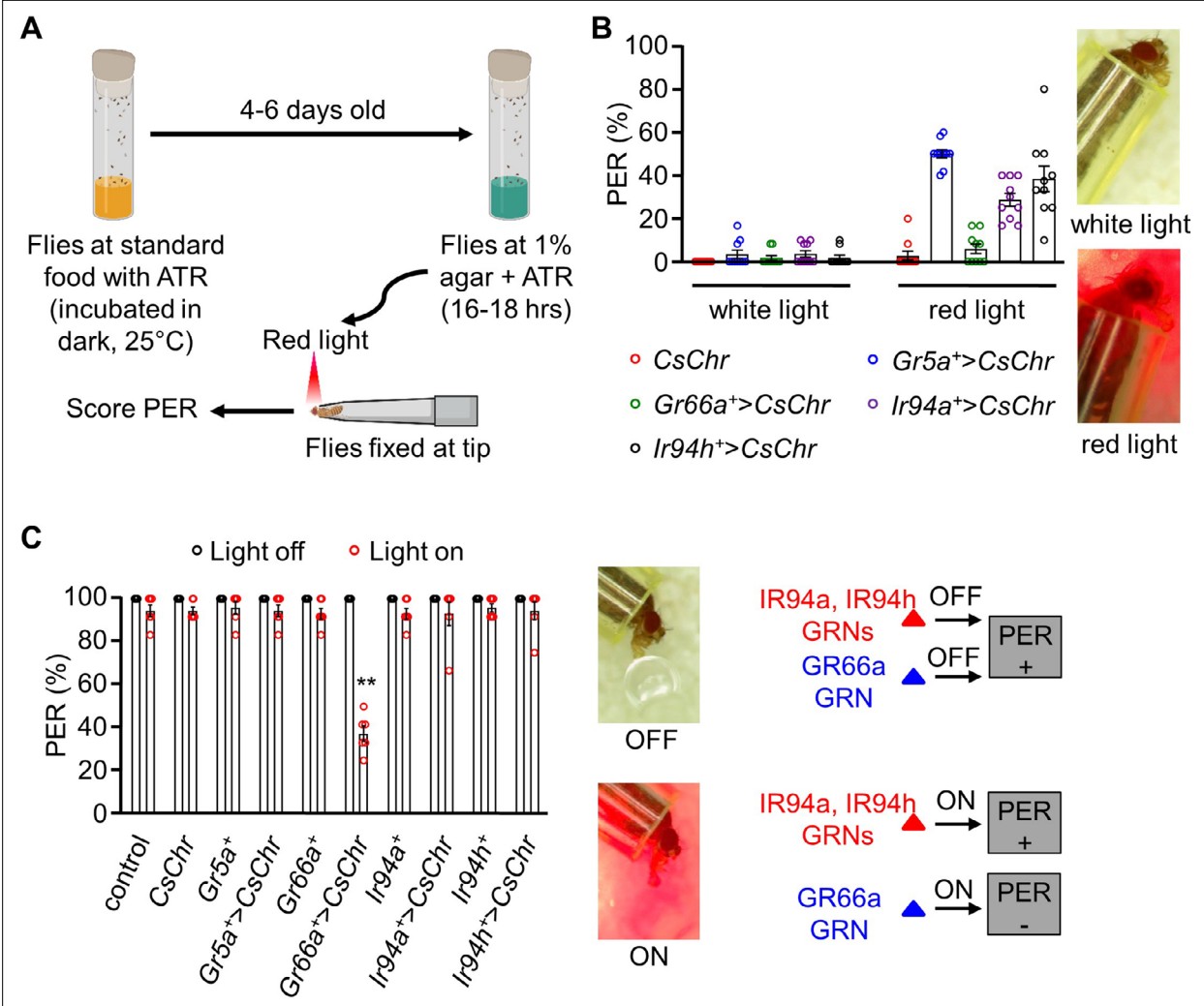

**Figure 5.** Stimulation of pharyngeal *Ir94a* and *Ir94h* gustatory receptor neurons (GRNs) using the optogenetic method. (**A**) Schematic representation of optogenetics methods performed in the study (details in the *Method* section). (**B**) Optogenetic activation of *Gr5a*, *Gr66a*, *Ir94a*, and *Ir94h* GRNs by expressing red-shifted channelrhodopsin (*UAS-CsChrimson*) under the control of respective *GAL4* drivers. White-light and red-light sources with low exposure time (3–5 s) were used in the experiment (n = 6–10 biological replicates). (**C**) Optogenetic activation of *Gr5a*, *Gr66a*, *Ir94a*, and *Ir94h* GRNs by expressing red-shifted channelrhodopsin (*UAS-CsChrimson*) under the control of respective *GAL4* drivers with sucrose stimulation in white-light and red-light source. Left: Photograph showing the proboscis extension response (PER) in present and absence of red-light source. Bottom: Schematic representation showing PER response to sucrose in presence or absence of red light (n = 6 biological replicates). The control flies were $w^{1118}$. All error bars represent SEMs. Single-factor ANOVA with Scheffe's analysis was used as a post hoc test to compare multiple sets of data. Different colored circular dots represent the number of trials performed. Black asterisks indicate statistical significance compared with control; **p < 0.01.

The online version of this article includes the following figure supplement(s) for figure 5:

**Figure supplement 1.** Stimulation of pharyngeal *Ir94a* and *Ir94h* gustatory receptor neurons (GRNs) via chemogenetic method.

We expanded our study to investigate if IR25a and IR76b receptors play a role in the pharyngeal response to carboxylic acids. Testing *Ir94a-GAL4* and *Ir94h-GAL4* in conjunction with $Ir25a^2$ and $Ir76b^1$ mutants, we found a reduced fluorescence intensity when exposed to 1% CA and 1% GA in *Ir94a* and *Ir94h* pharyngeal GRNs. While the reduction was not as severe as in $Ir94a^1$ and $Ir94h^1$ (**Figure 6F**), restoring the diminished fluorescence activity was achieved by activating the genes through pharyngeal-specific GRNs using cell type-specific expression analysis (**Figure 6F**). This suggests that IR25a and IR76b act as coreceptors in the *Ir94a*- and *Ir94h*-expressing GRNs, playing an essential role in pharyngeal GRN-mediated sour taste perception.

We also investigated the role of *Ir94a* and *Ir94h* GRNs in sensing other acids (LA and HCl), sucrose, and a bitter compound, quinine. However, we did not observe intense changes in fluorescence

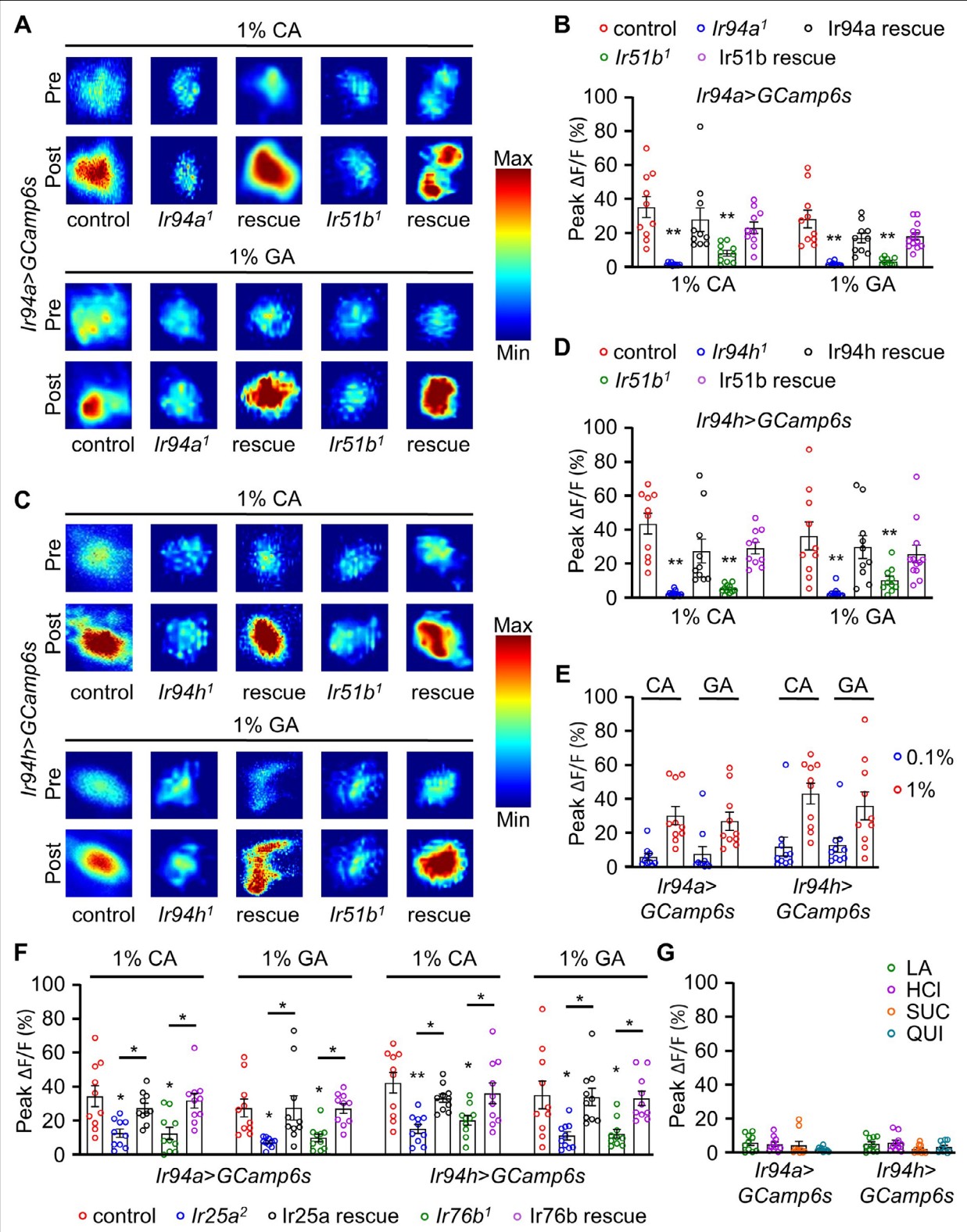

**Figure 6.** Calcium imaging analysis showing the activation of pharyngeal *Ir94a* and *Ir94h* gustatory receptor neurons (GRNs) with carboxylic acid applied as stimulus. (**A**) Heatmap analysis of pre- and post-stimulus with *Ir94a-GAL4* flies with expression of GCaMP6s in control, mutant (*Ir94a^1* and *Ir51b^1*) background, and rescue flies. (**B**) Quantification of peak fluorescence changes with stimulation by 1% citric acid (CA) and glycolic acid (GA) for *Ir94a-GAL4* flies with expression of GCaMP6s calcium indicator (*n* = 10–12 biological replicates). (**C**) Heatmap analysis of pre- and post-stimulus with *Ir94h-GAL4* flies with expression of GCaMP6s in control, mutant (*Ir94h^1* and *Ir51b^1*) background, and rescue flies. (**D**) Quantification of peak fluorescence

*Figure 6 continued on next page*

*Figure 6 continued*

changes with stimulation by 1% CA and GA for *Ir94h-GAL4* flies with expression of GCaMP6s calcium indicator (*n* = 10–12 biological replicates). (**E**) *Ir94a* and *Ir94h* neuron calcium imaging in 0.1% and 1% CA and GA (*n* = 10 biological replicates). (**F**) Quantification of peak fluorescence changes with stimulation by 1% CA and GA for *Ir94a-GAL4* and *Ir94h-GAL4* flies with expression of GCaMP6s calcium indicator in control, mutant (*Ir25a²* and *Ir76b¹*) background and rescue flies (*n* = 10 biological replicates). (**G**) Peak fluorescence level intensity stimulated by 1% lactic acid (LA), 1% hydrochloric acid (HCl), sucrose (SUC), and quinine (QUI) with *Ir94a-GAL4* and *Ir94h-GAL4* flies by expression of GCaMP6s calcium indicator (*n* = 10 biological replicates). All error bars represent SEMs. Single-factor ANOVA with Scheffe's analysis was used as a post hoc test to compare multiple sets of data. Colorful circle represents the number of trials performed. Black asterisks indicate statistical significance compared with control; *$p<0.05$, **$p < 0.01$.

The online version of this article includes the following figure supplement(s) for figure 6:

**Figure supplement 1.** Feeding assay analysis of control and mutants (**A-B**) .

**Figure supplement 2.** Recapitulation of sour receptors.

intensity after the application of these stimulants (*Figure 6G*). Both CA and GA induced a progressive increase in the GCaMP6s signal compared to the pre-stimulus fluorescence activity baseline, whereas other appetitive compounds did not enhance the response. In our previous research, we found that three IR mutants (*Ir51b¹*, *Ir94a¹*, and *Ir94h¹*) do not play a role in attraction to vitamin C, a sour tastant (*Shrestha et al., 2023a*). Furthermore, when tested for aversive response to caffeine and attractive response to salt (*Sang et al., 2024*), these mutants showed no significant differences in preference or avoidance compared to the control group when exposed to sucrose, caffeine, or salt (*Figure 6—figure supplement 1A*). This implies that IR51b, IR94a, and IR94h are specifically required for detecting carboxylic acids (CA and GA), although IR51b is also essential for sensing LA.

To delve deeper into the taste perception of the candidate IR mutants, we explored their role in detecting strong acids, specifically HCl, in addition to weak carboxylic acids. Using feeding assays with 1 and 100 mM HCl, we observed that the control group preferred the slightly lower concentration of HCl but avoided the higher concentration. Interestingly, the response of the candidate IR mutants mirrored that of the control group for both concentrations (*Figure 6—figure supplement 1B*). In summary, our findings suggest that IR25a, IR51b, IR76b, IR94a, and IR94h are responsible for the gustatory sensation of appealing concentrations of carboxylic acids and specifically contribute to the perception of sour taste rather than other taste modalities.

Moreover, we ectopically expressed *UAS-Ir51b*, *UAS-Ir94a*, and *UAS-Ir94h* under the guidance of *Gr33a-GAL4* and checked whether it elicits an enhanced taste-induced neuronal response from S- or I-type sensilla similar to L-type sensilla. On checking the neuronal firing from S-type (S3, S6, and S10) and I-type (I8, I9, and I10) sensilla of ectopically expressed flies, the response was similar to control flies (*Figure 6—figure supplement 2*). This indicates that these three IRs are not enough to recapitulate sour receptors. Overall, our current results, in combination with our previously published results, illustrate that peripheral sweet-sensing neurons and pharyngeal neurons are crucial requisites for carboxylic acid-sensing.

In conclusion, our findings from behavioral and physiological experiments demonstrate that *Ir94a* and *Ir94h* GRNs are specifically tuned to CA and GA, with IR51b acting as a coreceptor, along with IR25a and IR76b. Together, our results suggest that sour taste perception is accomplished through two parallel IR-mediated pathways in the gustatory system of *Drosophila*.

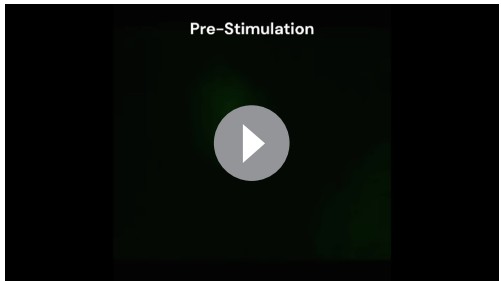

**Video 1.** Ex vivo calcium imaging of *Ir94a-GAL4* from ventral cibarial sense organ (VCSO) in pharynx with the stimulus 1% citric acid (CA).

https://elifesciences.org/articles/101439/figures#video1

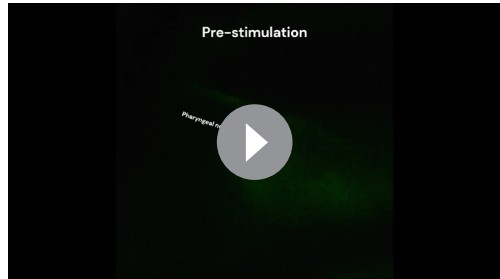

**Video 2.** Ex vivo calcium imaging of *Ir94h-GAL4* from labral sense organ (LSO) in pharynx with the stimulus 1% glycolic acid (GA).

https://elifesciences.org/articles/101439/figures#video2

## Discussion

Our study reveals the pharyngeal cellular mechanism by which the *Drosophila* gustatory system detects and evaluates sour taste stimuli. We identified specific pharyngeal neurons, namely IR94a and IR94h, as molecular sensors for carboxylic acids, with IR25a, IR51b, and IR76b acting as coreceptors. These findings highlight the importance of pharyngeal taste sensors in sour taste perception and provide insights into the complex processes that govern taste perception in insects.

Our research indicates that a group of IRs plays a crucial role in mediating the pleasant taste experience associated with optimal concentrations of carboxylic acids, with a specific emphasis on their involvement in perceiving sourness, as opposed to other taste qualities. In addition, we have previously reported that IR51b is essential for detecting aversive nitrogenous waste in the labellum (*Dhakal et al., 2021*). Therefore, IR51b functions as an aversive receptor in the labellum but as an attractive receptor in the pharynx. Thus, IR51b may not be specifically dedicated to sour taste perception but could potentially act as a coreceptor in the pharynx for detecting three carboxylic acids. Our attempts to identify IR51b expression using the GAL4 reporter failed due to the ineffectiveness of our knock-in-GAL4 (*Ir51b²*). Consequently, we were unable to identify the specific neurons responsible for detecting LA, and further research is needed to determine the responsive GRNs for LA. Another intriguing finding is the role of *Ir51b* function in either IR94a or IR94h neurons is sufficient for attraction raises questions about two separate circuits within the pharynx. Understanding the specific mechanisms behind this observation could provide insights into how different neurons collaborate to mediate behavioral responses to stimuli.

We revealed that both IR25a and IR76b serve as coreceptors within the GRNs expressed in the pharynx. Physiological examinations of mutants lacking IR25a and IR76b demonstrated noteworthy but partial reductions in the responsiveness of pharyngeal GRNs to CA and GA stimuli. This diminished responsiveness was effectively restored by cell type-specific expression through pharyngeal GRNs. The physiological responses in the rescued flies closely resembled those in control flies, suggesting that the defective property is intrinsic to individual cells. This intrinsic defect operates independently of external influences, emphasizing the autonomy of the cells and underscoring the role of pharyngeal receptors, including the identified coreceptors (IR25a and IR76b). Thus, we believe that pharyngeal cells have a predominant role in perceiving carboxylic acid taste.

We observed the differences in behavioral characteristics among the identified receptor mutant lines. For instance, while *Ir94a* mutants exhibit neutrality toward CA, *Ir94h* mutants strongly avoid it. We believe that while all the identified genes may be involved in the same sensory process, they likely have distinct functions within that process. Some genes may play a more critical role or exert a higher degree of influence on the overall sensation. This could explain the varying degrees of defect severity among the associated genes. Additionally, if an identified gene has a substantial number of crucial interactions, its mutation may have a cascading effect, resulting in a more severe defect compared to genes with fewer interactions or less critical partners. Therefore, each identified gene and its associated mutation should be evaluated within the specific sensory process and in the context of the overall genetic and biological framework to understand the variations in defect severity. Conceivably, these receptors could activate some other pathway through an independent cellular mechanism, indicating its importance not only in attraction to acid but also in masking the acid avoidance mechanism. Additionally, this discrepancy hints at the complex interplay between sensory neurons and the stimuli they detect. Reference to previous research highlighting the aversive nature of carboxylic acids at high concentrations adds depth to this observation, suggesting potential variations in dose–response dynamics mediated by these mutants in different way (*Ganguly et al., 2021*; *Rimal et al., 2019*).

The reception of different carboxylic acids achieved by different receptors in our study is compelling. We believe that it is a multifaceted process that involves a range of factors and mechanisms. This selectivity likely arises from a combination of structural specificity, with the receptors possessing unique binding sites or domains that interact with the distinct chemical structures of different carboxylic acids. The differences in chemical structure, size, shape, or properties that lead to differentiation in acidity can impact the receptors' ability to recognize and bind to them, resulting in different behavioral and neural responses. Moreover, taste neurons in *Drosophila* appear to possess broader dynamic ranges than previously anticipated. Additionally, they exhibit responsiveness to various chemicals even without direct contact and demonstrate OFF responses that correlate taste attributes with behavioral responses (*Dweck and Carlson, 2023*). Examining these factors is essential for comprehending how

taste circuits convert chemical information into behavior and for extending the understanding of the taste-based coding mechanism of specific ligand–receptor interactions. Furthermore, evolutionary adaptation may have contributed to the selectivity, with organisms developing receptors tailored to compounds relevant to their survival, such as those in their natural diet. More importantly, the presence of the $CH_3$ (methyl) group in LA, as opposed to GA or CA, may contribute to some of the differences in their properties. Receptors are highly specific in recognizing and binding to certain molecules, often having binding sites with complementary shapes and chemical properties to the molecules with which they interact. The methyl group, for example, contributes to the receptor specificity. In our investigation of various organic acids, we recognized the significance of the anion component. However, our findings also highlight the crucial role of each acid's acidity in determining its attractive properties (*Figure 2—figure supplement 1C*). Consequently, the unique specificity displayed by IR94a and IR94h receptors may also arise from their ability to discern the potent acidic characteristics and appealing flavors generated by specific ligands associated with CA and GA. These receptors appear finely tuned to recognize the intense acidity and pleasurable taste attributes conferred by these particular compounds, thereby elucidating their role in the sensory perception of flavors.

Based on our findings, we propose that labellar sour sensors are primarily involved in the initial detection of acidic tastes, influencing the extension response of the proboscis, whereas pharyngeal sour sensors are essential for the ingestion of acidic food. The labellar sensors appear to have minimal or no role in the ingestion process. Moreover, other research groups have demonstrated that *poxn* mutants still exhibit a response to sweet compounds (*Chen et al., 2021*; *LeDue et al., 2015*), suggesting the existence of additional internal pharyngeal neurons responsible for other taste modalities. It would be worthwhile to investigate if these internal pharyngeal neurons also mediate other taste modalities.

The activation of sweet-sensing GRNs by LA has been shown to have two distinct phases, as confirmed by calcium imaging in the brain (*Stanley et al., 2021*). IR25a mediates the onset response, and the removal phases are mediated by sweet GRs, resulting in elevated calcium responses upon the removal of LA. Our previous findings revealed that the fructose receptor GR43a is essential for the behavioral response to carboxylic acids; however, it does not operate in the labellum, indicating that the behavioral assay may capture effects associated with the post-ingestive roles of GRs (*Shrestha and Lee, 2021*). Similarly, it would be valuable to investigate the functional requirement of sour-sensing GRs in the pharynx. Additionally, recent observations have revealed relationships between the OFF response (but not the ON response) and tastant structure and behavior (*Dweck and Carlson, 2023*). Thus, it would be noteworthy to examine these relationships of various acidic tastants. Such an analysis could unveil further aspects of acidic taste coding.

Subsequent research endeavors should strive to elucidate the precise convergence of the two parallel pathways within the brain that collaboratively oversee sour taste perception. Delving deeper into this intricate interaction would shed light on the complex orchestration of sensory information. Furthermore, an exploration into the intricate neural circuits that establish connections between peripheral and pharyngeal gustatory neurons holds the promise of unearthing even more profound insights into the mechanisms at play. Additionally, the precise neural circuits involved in the integration of the sensory inputs and the behavioral outputs remain to be elucidated. Exploration of the downstream signaling mechanisms of pharyngeal GRNs and potential connection with other taste modalities is one of the interesting areas of research based on our study.

This study's exploration of the sensory biology of taste in *Drosophila* is a significant advancement in our understanding of gustatory mechanisms, providing a robust framework for future investigations into the sour taste perception among insects. By elucidating the complexities of nutrient selection and feeding strategies, this research not only enhances our comprehension of insect behavior but also bridges a gap to broader biological processes, including those in humans. The findings prompt a reevaluation of how organisms, from insects to humans, interact with their environments through molecular sensory pathways, highlighting the universal aspects of taste biology across diverse species. This cross-species insight is invaluable for developing more comprehensive models of sensory-driven behaviors in ecological and evolutionary contexts. Additionally, investigating the ecological significance of the lesser-studied pharyngeal neurons in *Drosophila*, which are key to the fly's perception of carboxylic acid taste, presents a valuable opportunity to deepen our understanding of sensation and mechanism behind it. These neurons are critical for essential survival behaviors, including feeding.

Research into the natural stimuli activating these neurons and their role in the fly's fitness could reveal their integration into the species' ecological and evolutionary strategies. Such studies could aid in our knowledge of taste perception and enlighten us on the adaptive importance of sensory systems in nature.

## Materials and methods

### *Drosophila* stocks

The wild-type control for all experiments was $w^{1118}$. Flies were maintained at 25°C under 12 hr light/12 hr dark cycles. Both male and female strains were used randomly for the experiments. We obtained the following lines from the Bloomington *Drosophila* Stock Center (https://bdsc.indiana.edu/): *Ir7g¹* (BL42420), *Ir8a¹* (BL41744), *Ir10a¹* (BL23842), *Ir21a¹* (BL10975), *Ir48a¹* (BL26453), *Ir48b¹* (BL23473), *Ir51b¹* (BL10046), *Ir52b¹* (BL25212), *Ir52c¹* (BL24580), *Ir56b¹* (BL27818), *Ir56d¹* (BL81249), *Ir62a¹* (BL32713), *Ir67a¹* (BL56583), *Ir75d¹* (BL24205), *Ir85a¹* (BL24590), *Ir92a¹* (BL23638), *Ir94b¹* (BL23424), *Ir94d¹* (BL33132), *Ir94f¹* (BL33095), *Ir94g¹* (BL25551), *Ir100a¹* (BL31853), *Ir94a-GAL4* (BL60720), *Ir94h-GAL4* (BL60728), *poxn⁷⁰* (BL60688), *UAS-mCD8-GFP* (BL5137), *UAS-hid* (BL65403), *UAS-Kir2.1* (BL6596), *Ir94a-c (Df)* (BDSC5598), and *Ir94d-h (Df)* (BDSC7673). We described the source of the following lines in our previous study: *Ir7a¹*, *Ir47a¹*, *Ir52a¹*, *Ir56a¹*, *Ir60b³*, *Ir94c¹*, and *UAS-Ir25a* (**Lee et al., 2018**; **Rimal et al., 2019**). L. Vosshall and G.B. Suh provided *Ir25a-GAL4*, *Ir25a²* (**Benton et al., 2009**), and *poxn^{ΔM22−B5}*. C. Montell kindly provided *UAS-GCaMP6s*, *Ir76b¹* (**Zhang et al., 2013**), and *Gr33a-GAL4* (**Moon et al., 2009**). K. Scott provided *ppk23-GAL4* (**Thistle et al., 2012**), and H. Amrein provided *Gr5a-GAL4* (**Thorne et al., 2004**). A. Dahanukar contributed *Ir20a¹* and *Gr64f-GAL4* (**Dahanukar et al., 2007**). M. Gordon provided *Ir7c^{GAL4}* (**McDowell et al., 2022**). *UAS-CsChrimson* was received from the Korea Drosophila Research Center (http://kdrc.kr/index.php). *UAS-Ir51b* and *Ir51b²* have been described earlier (**Dhakal et al., 2021**). *UAS-Ir94a*, *UAS-Ir94h*, *Ir94a¹*, and *Ir94h¹* were generated for the purpose of this study. The aforementioned strains were employed in our studies to examine the precise impacts and functions of the appropriate genes or proteins linked to each strain.

### Chemical reagents

GA (Cat# 124737), CA (Cat# 251275), LA (Cat# 252476), tricholine citrate (TCC; Cat# T0252), sucrose (Cat# S9378), quinine hydrochloride dihydrate (Cat# Q1125), caffeine (Cat# C0750), sulforhodamine B (Cat# 230162), capsaicin (Cat# M2028), and KOH (Cat# 306568) were purchased from Sigma-Aldrich Co. HCl (Cat# 7647-01-0) was purchased from Samchun Chemical Co. Ltd. Brilliant blue FCF (Cat# 027-12842) was purchased from Wako Pure Chemical Industry Ltd.

### Generation of *Ir94a¹*, *Ir94h¹*, *UAS-Ir94a*, and *UAS-Ir94h* lines

The *Ir94a¹* and *Ir94h¹* mutants were generated by ends-out homologous recombination (**Gong and Golic, 2003**). To generate the construct intended for injection, the two 3 kb genomic fragments were amplified by PCR (*Not*I and *Bam*HI were used for both), and the DNA was subcloned into the pw35 vector. Assuming the 'A' of the 'ATG' start codon as '+1', the deleted region was −199 to +653 for *Ir94a¹* and −117 to +547 for *Ir94h¹*. The construct was injected into $w^{1118}$ embryos by Best Gene, Inc. We outcrossed the mutant with $w^{1118}$ for six generations.

To create the *UAS-Ir94a* and *UAS-Ir94h* transgenic strain, mRNA was used in RT-PCR to amplify the full-length *Ir94a* and *Ir94h* cDNA, which were then subcloned into the pUAST vector between the *Not*I and *Xba*I for both *UAS-Ir94a* and *UAS-Ir94h* sites, respectively. The following primer set was used for amplification:

> For *UAS-Ir94a*:
> Forward primer (*Not*I): 5′-TTGCGGCCGCTAGAAAATGGCTTTA-3′
> Reverse primer (*Xba*I): 5′-TATCTAGATTTAAGCATTAGAC-3′
> For *UAS-Ir94h*:
> Forward primer (*Not*I): 5′-AGCGGCCGCATGTTGTCCAACATCAGT-3′
> Reverse primer (*Xba*I): 5′-TTCTAGACTACAAGCTATTGAGAAA-3′.

The cloned cDNA was confirmed through DNA sequencing. The transformation vector containing the respective constructs was injected into $w^{1118}$ embryos (KDRC).

## Binary food choice assay

We conducted binary food choice assays, as described previously (*Aryal et al., 2022b*). Around 50–70 flies (4–6 days old; mixed sexes) were starved by incubating for 18 hr in a vial (50–60% relative humidity, 25°C, 12/12 hr light/dark cycle) containing 1% agarose. Two different food sources containing 1% agarose were prepared: one containing 2 mM sucrose and the other containing 2 mM sucrose with different concentrations of acid or other tastants. Another two foods containing 1 mM sucrose alone or 5 mM sucrose with different tastants were prepared for a few experiments. In one experiment, the pH of 1% LA, 1% CA, and 1% GA with 2 mM sucrose was increased with the addition of KOH. These food sources were mixed with either blue food coloring (brilliant blue FCF, 0.125 mg/ml) or red food coloring (sulforhodamine B, 0.1 mg/ml). The two mixtures were distributed in an alternating manner in the wells of a 72-well microtiter dish (Thermo Fisher Scientific, Cat# 438733). Within 30 min of preparation, previously starved flies were introduced to the dish. The dish was then incubated in a dark, humidified chamber, and the flies were allowed to feed for 1.5 hr at 25°C. The dish was transferred to −20°C to sacrifice the flies. The color of their abdomens was checked for analysis under a stereomicroscope. Blue ($N_B$), red ($N_R$), or purple ($N_P$) flies were tabulated. The preference index (P.I.) was calculated according to the following equation: $(N_B − N_R)/(N_R + N_B + N_P)$ or $(N_R − N_B)/(N_R + N_B + N_P)$, depending on the dye/tastant combinations. P.I. = 1.0 and −1.0 indicate complete preferences for one of the other foods, and P.I. = 0.0 indicates no bias between the two food choices.

## Electrophysiology

Electrophysiology was conducted, as previously described (*Lee et al., 2009*). At first, 4- to 6-day-old flies were anesthetized on ice. Then, a reference glass electrode filled with Ringer's solution was inserted into the thorax of the flies, extending the electrode toward their proboscis. To avoid any possible biases, five to six live insects were prepared per setup, and the same procedure was repeated for several rounds on different days. The sensilla were stimulated for 5 s with different tastants dissolved in a 30-mM TCC solution (i.e., electrolytes) in recording pipettes (10–20 µm tip diameter). The recording electrode was connected to a preamplifier (Taste PROBE, Syntech), and the signals were collected and amplified tenfold using a signal connection interface box (Syntech) in conjunction with a 100- to 3000-Hz band-pass filter. Recordings of action potentials were acquired using a 12-kHz sampling rate and analyzed using the AutoSpike 3.1 software (Syntech). Each consecutive recording was performed within approximately 1 min between each stimulation. The black circles in each figure indicate the number of insects tested.

## Immunohistochemistry

This method was conducted as described earlier (*Lee et al., 2018*). The labella of flies were dissected and fixed using 4% paraformaldehyde (Electron Microscopy Sciences, Cat# 15710) in PBS-T (1× phosphate-buffered saline and 0.2% Triton X-100) for 15 min at room temperature. The fixed tissues were washed carefully three times with PBS-T. After washing, tissues were cut in half with a razor blade. The samples were then incubated for 30 min at room temperature in blocking buffer (0.5% goat serum in 1× PBS-T). The primary antibodies (1:1000 dilution; mouse anti-GFP [Molecular Probes, Cat# A11120]) were added to freshly prepared blocking buffer and incubated with the samples overnight at 4°C. The samples were washed three times with PBS-T at 4°C and incubated with secondary antibodies (1:200 dilution in the blocking buffer; goat anti-mouse Alexa488 [Thermo Fisher, Cat# A11029]) for 4 hr at 4°C. The tissues were then washed three times with PBS-T and placed in mounting buffer (37.5% glycerol, 187.5 mM NaCl, 62.5 mM Tris, pH 8.8) and viewed using an inverted Leica LASX confocal microscope.

## Ex vivo calcium imaging

Ex vivo calcium imaging was performed using a low-melting agarose method. For the experimental process, we used 4- to 7-day-old flies expressing *UAS-GCaMP6s* driven by *Ir94a-GAL4* and *Ir94h-GAL4* (incubation condition: 50–60% relative humidity, 25°C, 12/12 hr light/dark cycle). For the experiment, 0.5% low-melting agarose was prepared and distributed on the confocal dish (SPL Life Sciences, Cat# 102350). The shallow well with a blade was prepared for sample fixation. Next, the head of each fly was decapitated carefully using sharp razor blades, followed by the excision of the slight part of the labellum of the extended region of the proboscis for easy access of tastant to pharyngeal

organs (LSO for *Ir94h-GAL4* and VCSO for *Ir94a-GAL4*). The prepared tissue sample was later carefully fixed in an inverted position on the well already prepared earlier. Adult hemolymph solution (108 mM NaCl, 5 mM KCl, 8.2 mM MgCl$_2$, 2 mM CaCl$_2$, 4 mM NaHCO$_3$, 1 mM NaH$_2$PO$_4$, 5 mM HEPES (4-(2-hydroxyethyl)piperazine-1-ethanesulfonic acid), pH 7.5) was kept as the pre-stimulus solution, which was followed by the stimulus solution after 30 s. Stimulus solution was inserted slowly on pre-stimulus solution to avoid any kind of liquid phase movement. This allowed direct access of the stimulus to the pharyngeal neurons. GCaMP6s fluorescence was viewed with a fluorescence microscope. The relevant area of the pharynx was visualized using the 10×objective. A video was recorded at a speed of two frames scanned per second. Changes in neuronal fluorescent activity were recorded for 4 min after supplying the stimulus. Throughout the experiment, following the stimulation, the tissue remained exposed to the chemical due to the absence of mechanisms for its removal. Typically, the response onset occurred within 1–2 min following the stimulation period. Fluorescence intensities (*F*) were measured using Fiji/ImageJ software (https://fiji.sc). A region of interest (ROI) was drawn around the cell bodies. For the analysis, the average intensity for ROIs during each frame was measured using the Time-Series Analyzer Plugin written by J. Balaji (https://imagej.nih.gov/ij/plugins/time-series.html). If there was modest movement of sample, it was measured manually at different fraction of time in same ROI and subsequently merged. The change in fluorescence intensity (Δ*F*) relative to the baseline fluorescence (*F*) represented by Δ*F*/*F* (%) was calculated as ($F_{max}$ − $F_0$)/$F_0$ × 100%. $F_0$ is the GCaMP6s baseline value averaged for 10 frames immediately before stimulus application. $F_{max}$ is the maximum fluorescence value following stimulus delivery. Each Δ*F*/*F* (%) was presented as dot plot in respective figures.

## Optogenetic assay

Two different optogenetics methods were used in the study. Flies were grown on standard cornmeal agar supplemented with 400 µM ATR (Sigma-Aldrich, Cat# R2500) at 25°C under dark conditions. After eclosion, 4- to 6-day-old flies were kept in a vial containing 1% agarose with ATR for 18–24 hr under the same conditions. The next day, each fly was trapped within a trimmed 200 µl pipette tip so that the proboscis was exposed to the outside while the remainder of the fly was secured inside the pipette tip. In the first method, the optogenetic experiment was performed as described previously with a few modifications (*Joseph et al., 2017*). To the above-described flies under the controlled condition, the fly proboscis was stimulated with red light or white light. Those with full or partial proboscis extension were recorded as positive. The light source was activated for 3–5 s for respective experiments. In the second method, we stimulated the fly proboscis with sucrose in the initial phase, and those showing a positive response were stimulated using sucrose, followed by red light. The PER was recorded as full and partial. In each trial (*n*), flies showing a positive response to sucrose were used for further experiments. The PER percentage was calculated by the percentage of the total number of positive responses to red lights.

## RT-PCR analyses

The whole body, labellum, and pharynx samples were dissected from control (*w$^{1118}$*) adult flies for *Figure 3E*. Only the pharynx was dissected from control and *poxn* null alleles (*poxn$^{70}$* and *poxn$^{ΔM22−B5}$*) for *Figure 3F*. Achieving a pristine dissection of the pharynx necessitated the application of meticulous and precise techniques, coupled with diligent handling and preparation. Initially, the dissection commenced with the removal of the labellum section using a sterile surgical blade, executed with utmost care to minimize any inadvertent damage. Subsequently, employing another sterile surgical blade, the targeted region of the pharynx was delicately isolated from the surrounding proboscis tissue. This isolated pharyngeal segment was then meticulously collected, ensuring its integrity and purity for subsequent experimental analyses. Total RNA was extracted using TRIzol (Invitrogen), and cDNA was synthesized using AMV reverse transcriptase (Promega). To perform the RT-PCR, we used the following primers:

> *Ir51b* forward primer: 5'-GGCGCTAACAAACGCTGCTTAC-3'
> *Ir51b* reverse primer: 5'-CAGAGCTGACAGTATCCAACCAA-3'
> *tubulin* forward primer: 5'-TCCTTGTCGCGTGTGAAACA-3'
> *tubulin* reverse primer: 5'-CCGAACGAGTGGAAGATGAG-3'.

The RT-PCR reactions were run for 35 cycles to amplify the target gene fragments. The resulting RT-PCR products were subsequently analyzed to determine gene expression levels. To ensure the integrity and purity of the RNA preparation and the absence of genomic DNA contamination, we included a No Reverse Transcriptase (no-RT) sample as a control, which consisted of RNA without the treatment of AMV reverse transcriptase enzymes before proceeding to PCR.

## Quantification and statistical analysis

The experiments were performed on several different days, and the data were analyzed using GraphPad Prism (ver. 8.0; Dotmatics). Raw values are illustrated on the graph plots. All the sample sizes are stated in the corresponding figure legends. All statistical analyses were performed using Origin Pro 8 for Windows (ver. 8.0932; Origin Lab Corp.). All error bars represent SEMs. Single-factor ANOVA coupled with Scheffe's post hoc test was conducted to compare multiple sets of data, and unpaired Student $t$-tests were used for comparisons within datasets. Black asterisks indicate statistical significance from the control and each mutant, respectively ($p < 0.05$, $p < 0.01$).

## Acknowledgements

We thank Dr. C Montell, Dr. LB Vosshall, Dr. K Scott, Dr. A Dahanukar, Dr. GB Suh, Dr. H Amrein, Dr. SJ Moon, and Dr. M Gordon for kindly providing fly reagents. We are also grateful to Bloomington *Drosophila* Resource Center and Korea Drosophila Resource Center for providing fly stocks. This work was supported by grants to YL from the National Research Foundation of Korea (NRF) funded by the Korea government (MIST) (NRF-2021R1A2C1007628), and the Biomaterials Specialized Graduate Program through the Korea Environmental Industry and Technology Institute (KEITI) funded by the Ministry of Environment (MOE). BS and SR were supported by the Global Scholarship Program for Foreign Graduate Students at Kookmin University in Korea.

## Additional information

### Funding

| Funder | Grant reference number | Author |
| --- | --- | --- |
| National Research Foundation of Korea | NRF-2021R1A2C1007628 | Youngseok Lee |
| Korea Environmental Industry and Technology Institute | Biomaterials Specialized Graduate Program | Youngseok Lee |

The funders had no role in study design, data collection, and interpretation, or the decision to submit the work for publication.

### Author contributions

Bhanu Shrestha, Conceptualization, Resources, Data curation, Formal analysis, Investigation, Writing – original draft, Writing – review and editing; Jiun Sang, Resources, Investigation; Suman Rimal, Conceptualization, Formal analysis, Investigation; Youngseok Lee, Conceptualization, Supervision, Funding acquisition, Writing – original draft, Project administration, Writing – review and editing

### Author ORCIDs

Bhanu Shrestha  http://orcid.org/0000-0002-2945-2305
Youngseok Lee  https://orcid.org/0000-0003-0459-1138

### Ethics

Drosophila melanogaster as an LMO was handled in authorized facility.

Reviewer #1 (Public review): https://doi.org/10.7554/eLife.101439.2.sa1
Reviewer #2 (Public review): https://doi.org/10.7554/eLife.101439.2.sa2
Reviewer #3 (Public review): https://doi.org/10.7554/eLife.101439.2.sa3

Author response https://doi.org/10.7554/eLife.101439.2.sa4

## Additional files

### Supplementary files

- Supplementary file 1. Statistics for the data shown in *Figure 2—figure supplement 1A*.
- Supplementary file 2. Statistics for the data shown in *Figure 2—figure supplement 1B*.
- Supplementary file 3. Statistics for the data shown in *Figure 2—figure supplement 1C*.
- MDAR checklist

### Data availability

Source data for all figures contained in the manuscript and SI have been deposited in 'figshare' (https://doi.org/10.6084/m9.figshare.27857658.v1).

The following dataset was generated:

| Author(s) | Year | Dataset title | Dataset URL | Database and Identifier |
|---|---|---|---|---|
| Shrestha B, Sang J, Rimal S, Lee Y | 2024 | Pharyngeal neuronal mechanisms governing sour taste perception in *Drosophila melanogaster* | https://doi.org/10.6084/m9.figshare.27857658.v1 | figshare, 10.6084/m9.figshare.27857658.v1 |

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
